# Yeast Interactions and Molecular Mechanisms in Wine Fermentation: A Comprehensive Review

**DOI:** 10.3390/ijms22147754

**Published:** 2021-07-20

**Authors:** Francesca Comitini, Alice Agarbati, Laura Canonico, Maurizio Ciani

**Affiliations:** Dipartimento di Scienze della Vita e dell’Ambiente (DiSVA), Università Politecnica delle Marche, Via Brecce Bianche, 60131 Ancona, Italy; a.agarbati@univpm.it (A.A.); l.canonico@univpm.it (L.C.); m.ciani@univpm.it (M.C.)

**Keywords:** multistarter fermentation, yeast interactions, molecular mechanisms, non-*Saccharomyces* yeasts

## Abstract

Wine can be defined as a complex microbial ecosystem, where different microorganisms interact in the function of different biotic and abiotic factors. During natural fermentation, the effect of unpredictable interactions between microorganisms and environmental factors leads to the establishment of a complex and stable microbiota that will define the kinetics of the process and the final product. Controlled multistarter fermentation represents a microbial approach to achieve the dual purpose of having a less risky process and a distinctive final product. Indeed, the interactions evolved between microbial consortium members strongly modulate the final sensorial properties of the wine. Therefore, in well-managed mixed fermentations, the knowledge of molecular mechanisms on the basis of yeast interactions, in a well-defined ecological niche, becomes fundamental to control the winemaking process, representing a tool to achieve such objectives. In the present work, the recent development on the molecular and metabolic interactions between non-*Saccharomyces* and *Saccharomyces* yeasts in wine fermentation was reviewed. A particular focus will be reserved on molecular studies regarding the role of nutrients, the production of the main byproducts and volatile compounds, ethanol reduction, and antagonistic actions for biological control in mixed fermentations.

## 1. Introduction

Nowadays, the use of commercial *Saccharomyces cerevisiae* strains in wine production is a common practice. In the past, wine was produced through spontaneous fermentation conducted by microbiota naturally colonizing grapes and winery. In this way, many yeast strains and species contributed to wine fermentation determining unpredictable interactions, sometimes resulting in failure. For these reasons, the practice of controlled fermentation widespread since the 1970s, through the inoculum of pure *S. cerevisiae* starter cultures, quickly established a predictable process with a dominant yeast population carrying out well-managed, certain, and reliable results [1]. If the advantages correlated to pure wine fermentation such as the process management and quality of the final wine, the massive and widespread use of commercial *S. cerevisiae* strains reduce the microbial biodiversity of the process with a consequent reduction in wine complexity [2]. Indeed, the sensory profile of wines produced by monoculture-inoculated fermentations differs substantially from wines spontaneously fermented. This is demonstrated by the comparative study of the biochemical characteristics of wines inoculated with those from spontaneous fermentations, which are distinctly different [3]. Certainly, spontaneous fermentations imply a higher risk of sluggish and/or stuck fermentation, spoilage contamination, and a relatively undesired final aroma when compared to pure processes characterized by many defined and desirable characteristics but less complex flavor profiles [4].

For this reason, in the last few years, wine researchers have explored the controlled use of non-*Saccharomyces* starter cultures in addition to commercial *S. cerevisiae* strains [5,6]. It is certainly known that non-*Saccharomyces* yeasts are generally unable to complete alcoholic fermentation. For this, they are used in pairs with conventional starters. The goal to take advantage of multistarter fermentation, avoiding the risks of stuck fermentations, is reached by setting the conditions for which all participating yeasts cohabit in a stable way. This can be achieved by inoculating the two starter strains in coculture (non-*Saccharomyces* and *S. cerevisiae* strains) or first the non-*Saccharomyces* yeast followed by *S. cerevisiae* to finish the fermentation. This last modality is known as sequential inoculation [7,8]. During recent years, the controlled use of non-*Saccharomyces* yeasts in winemaking has grown enormously, quickly becoming a biotechnological tool to better understand the impact of the multistarter process on the chemical and sensorial properties of wine [9,10]. In this regard, analyzing comprehensive studies on controlled mixed fermentations in wine, two aspects emerged: (i) the wide intraspecific variability of non-*Saccharomyces* yeasts for the oenological characters; (ii) their different and specific behavior in coculture due to molecular interactions with *S. cerevisiae.* Experimental research based on these aspects strongly highlighted a significant role in the focus and application of non-*Saccharomyces* yeasts, determining the effect on the analytical and sensory profile and the aromatic complexity of wine. Indeed, it is well established that there are several specific purposes for the use of mixed cultures in fermented beverage production such as the improvement of specific wine traits, the enhancement of the complexity or peculiar structural/aromatic features of the final product, ethanol reduction, or the control of spontaneous or undesired microorganisms. In this context, the planned choice of the strain and the study of yeast interactions play an important role in the achievement of the desired features influencing the growth and/or metabolic pathway of mixed cultures. Moreover, a driving force of the market supported by a great demand of consumers is continuously increasing the interest in yeast–yeast interaction in mixed fermentation to provide biotechnological solutions to improve sensory characteristics [7].

Further efforts are needed to understand the modalities of such interactions and how each species contributes to fermentation. However, the available knowledge already defines the potentiality of well-managed mixed fermentations for the control of undesirable or spoilage microorganisms, in view of a sustainable perspective of organic wine production. Indeed, with countless possible yeast combinations, the potential of mixed fermentation in natural product discovery seems quite promising.

Here we reviewed the recent development in the studies on the molecular and metabolic interactions between non-*Saccharomyces* and *Saccharomyces* yeasts in wine fermentation. After a brief overview of the methods used for the molecular study, we examined the recent developments in the metabolic regulation in wine yeast interactions focusing on nutrients uptake, byproducts, and volatile compounds production, as well as ethanol reduction and antagonistic actions for biological control.

## 2. Methodological Approaches for the Study of Metabolic Interactions

Metabolomics is a branch of biochemistry involving comprehensive study on metabolites, bioactive compounds, or products of microbial metabolism [11]. In wine microbiology, there are few metabolomic studies, mainly focused on the evaluation of primary and secondary metabolites, their concentrations, and their relationships on the quality of wine. The most recent analytical techniques developed for metabolomics studies on wine allow the screening of hundreds of yeast metabolites using high-throughput methods [12].

The possibility to use a metabolomics approach in the study of the metabolic interactions between yeasts during mixed fermentations greatly increases the probability of understanding the molecular mechanisms underlying the interactions. For example, data obtained from Fourier transform ion cyclotron resonance mass spectrometry (FT-ICR-MS) and/or liquid chromatography coupled with tandem mass spectrometry (LC-MS/MS) analyses represent a useful metabolic footprint that discriminate wines based on the inoculated yeasts (in single or mixed culture) but also reveals differences related to the final aroma, providing a metabolomic picture of the wine [13].

Moreover, ultra-high-resolution mass spectrometry (uHRMS) is another tool to analyze the yeast metabolome after alcoholic fermentation. Roullier-Gall et al. [13] reported the change in wine chemical composition from pure and mixed-culture fermentation with *Lachancea thermotolerans*, *Starmerella bacillaris*, *Metschnikowia pulcherrima*, and *S. cerevisiae*. The authors, using this methodology, clearly differentiated wines according to pure or mixed fermentation. Again, uHRMS confirmed that cell–cell contact influences the metabolism of *L. thermotolerans* and *S. cerevisiae* [14]. For these reasons, metabolomics seems to be a suitable tool to better understand the microbial interaction during multistarter fermentation.

Interactions between different mixed cultures of yeasts studied by gas-chromatograph analysis demonstrated that wines produced from pure cultures had a different composition of volatile compounds compared to wines produced by cocultivation [15]. Also, NMR-based metabolomics was recently used to identify metabolites that discriminate between single and mixed cultures of two yeast during fermentation [16].

Other “omics” technologies were used to discover new relevant aspects of microbial interactions [13]. Transcriptomic analysis combined with physiological data can provide an integrated view into the response of a yeast to the environment during mixed-culture fermentation [17]. For instance, the available nutrients such as nitrogen and vitamins represent a factor that may determine population dynamics, fermentative activity, and byproducts formation.

## 3. Nutrient Uptake and Metabolic Response in Yeast-Yeast Interactions

Several biotic and abiotic mechanisms influence the yeast-yeast interactions during wine fermentation, determining, in a natural process, the microbial dynamics and the final dominance of one or few yeast strains. Although a wide variability, both in terms of quality and quantity, of non-*Saccharomyces* yeasts derived from grapes is present at the beginning of the process, *S. cerevisiae* is always the dominant species of fermentation. It has been well established that the capacity of *S. cerevisiae* strains to quickly consume the nitrogen and carbon sources available in grape juice, together with the high ethanol tolerance, gives them a key adaptive advantage over other yeasts [18]. Moreover, the abiotic stressful conditions during alcoholic wine fermentation are also important factors that positively impact *S. cerevisiae* dominance at the expense of non-*Saccharomyces* species. The availability and the nature of yeast assimilable nitrogen (YAN) compounds is a key factor for process management because, unlike other factors, it can be easily controlled by adding different organic and/or inorganic sources. Indeed, nitrogen is an essential nutrient during wine fermentation that increases biomass production and stimulates the rate of sugar utilization, while its deficiency can cause stuck or sluggish fermentations.

The relationship established between yeasts and nutrients is a very complex issue, even more so in a multispecies system such as wine fermentation [19]. Currently, little data on nitrogen needs, sources, and preferences are available among mixed fermentations [20], probably due to a wide variety of matrix parameters, yeast couples, and culture conditions.

Gobert et al. [21] studied the specific use uptake of amino acids and ammonium by three non-*Saccharomyces* yeast strains belonging to *S. bacillaris*, *M. pulcherrima*, and *Pichia membranifaciens* species in mixed fermentation with *S. cerevisiae*. The analysis of the YAN in mixed cultures showed that non-*Saccharomyces* yeasts have specific amino acid consumption profiles: histidine, methionine, threonine, and tyrosine were not consumed by *S. bacillaris*; aspartic acid was assimilated very slowly by *M. pulcherrima*; glutamine was not assimilated by *P. membranifaciens*. Differently, cysteine seems to be a preferred nitrogen source for all non-*Saccharomyces* yeasts. In sequential fermentation, these specific profiles of amino acid consumption by non-*Saccharomyces* yeasts may account for some of the interactions such as reduced performances of *S. cerevisiae* and volatile profile changes.

A possible explanation of specific non-*Saccharomyces* nitrogen uptake modality in a multispecies context could be their ability to change the transcriptional behavior in response to nutrient availability when compared with a single-species context. Using the transcriptomic approach, Barbosa and coauthors [17] showed that *S. cerevisiae* reduced its global transcription activity in coinoculation with *Hanseniaspora guilliermondii*. In particular, genes related to the biosynthesis of vitamins were upregulated, while genes involved in the uptake and biosynthesis of amino acids were downregulated. In another mixed fermentation model composed by *S. cerevisiae*/*Hanseniaspora uvarum* and *S. cerevisiae*/*Candida sake*, the results of the transcriptomic analysis showed a partial relief of nitrogen catabolite repression in *S. cerevisiae* as metabolic stimulation [22]. In a more recent work [23], a transcriptomic analysis of RNA-seq analysis demonstrated that *S. cerevisiae* in mixed fermentation with *Torulaspora delbrueckii* reduced the ammonium effects during fermentation. This behavior was already observed by Barbosa and coworkers in the *H. guillermondii*/*S. cerevisiae* system [17].

With the same intent, Lleixà et al. [24], analyzing the nitrogen consumption, together with the catabolism repression, demonstrated the positive involvement of *Hanseniaspora vineae* to produce wine with improved fermentation capacity and increased aromatic properties. This study also contributed to clarifying the controversial role of this apiculate yeast. Although it has been demonstrated that *H. vineae* increases the fruity aromas, producing high amounts of acetate esters (primarily 2-phenylethyl acetate), this yeast represents a concern in the fermentation of mixed cultures due to its consumption of nutrients, particularly nitrogen. However, these mechanisms of regulation and consumption are unclear. In this study, the use of synthetic must with standardized nitrogen content demonstrated a similar behavior of nitrogen regulation in *H. vineae* and *S. cerevisiae*, indicating the presence of specific nitrogen catabolism repression mechanism in the non-*Saccharomyces* yeast. In this case, the model of mixed fermentation contributed to the better understanding of nitrogen metabolism in an unexplored non-*Saccharomyces* species.

Recently, Su et al. [25] expanded the knowledge about the preferences and consumption rates of individual nitrogen sources by some non-*Saccharomyces* yeasts (*T. delbrueckii*, *M. pulcherrima*, and *Metschnikowia fructicola*) in a wine environment. They evidenced that during alcoholic fermentation, the non-*Saccharomyces* strains consumed different nitrogen sources in a similar order as *S. cerevisiae*, but not as quickly. Furthermore, when all the nitrogen sources were supplied in the same amount, their assimilation order was similarly affected for both *S. cerevisiae* and non-*Saccharomyces* strains. Another recent work confirmed that the management of the yeast assimilable nitrogen may be a powerful tool to modulate wine aroma profiles in mixed wine fermentation [26]. However, also from this last study emerged that further investigations should be carried out to clarify the metabolic fluxes of amino acid metabolism and volatile compounds production.

In addition to nitrogen metabolism, non-*Saccharomyces*/*S. cerevisiae* mixed fermentation may affect carbon-related byproducts, such as acetic acid (Figure 1). In this regard, several studies on different non-*Saccharomyces* species in sequential fermentation with *S. cerevisiae* highlighted their effect on the capacity to produce wines with low volatile acidity [27,28,29].

The competition for nutrients and metabolic resources during mixed fermentation also concerns the uptake and the metabolic flux of carbon sources. The evaluation of specific phenotypic traits of *S. bacillaris* in relation to central carbon metabolite and nitrogen sources showed that the non-*Saccharomyces* yeast exhibited low activity through the acetaldehyde pathway, which triggers an important redistribution of fluxes through the central carbon metabolic network [30]. In particular, the formation of metabolites derived from the two glycolytic intermediates glyceraldehyde-3-phosphate and pyruvate is substantially increased during fermentation by *S. bacillaris*. Effectively, the low production of ethanol and acetic acid by some non-*Saccharomyces* in general and by *S. bacillaris* in particular is strictly linked to the low activity of the acetaldehyde pathway. It is already known that this behavior has large-scale effects on the metabolic fluxes, requiring increased production of glycerol to overcome the lower production of ethanol and to maintain the redox balance of cells [31]. Furthermore, Englezos and coworkers [32] again reinforced the evidence of a metabolic reorientation of fluxes around the pyruvic acid and glyceraldehyde-3-phosphate nodes as a consequence of reduced carbon channeling toward the acetaldehyde pathway always in the *S. bacillaris* model. As a direct consequence, increased production of pyruvate and amino acids and larger amounts of alcohols derived from alanine, leucine, valine, and isobutanol, as well as metabolites from glyceraldehyde-3-phosphate, are shown.

The gene expression involved in pyruvate dehydrogenase bypass and glycerol pyruvic fermentation was already investigated by Sadoudi et al. [33] to monitor the acetic acid and glycerol production in *S. cerevisiae* and the sequential formation of *M. pulcherrima* in Sauvignon Blanc. In this case, *S. cerevisiae* exhibited a high expression of pyruvate decarboxylase PDC1 and PDC5, acetaldehyde dehydrogenase ALD6, alcohol dehydrogenase ADH1, and glycerol-3-phosphate dehydrogenase PDC1 genes during the first 3 days of fermentation in sequential fermentation. These results highlighted that the metabolic pathway of these two fermentation products can be affected by the presence of *M. pulcherrima,* determining an increase in glycerol content and a decrease in acetic acid.

In a comparative transcriptomic study carried out on an *S. cerevisiae* wine strain and *Saccharomyces kudriavzevii*, del Real and coworkers [34] demonstrated that in mixed fermentation, *S. cerevisiae* accelerated the nutrient uptake and utilization to outcompete the coinoculated yeast through a cell-to-cell contact mechanism. It was seen that *S. kudriavzevii* exhibited a specific response to competition that involved carbon and nitrogen nutrient uptake.

Another recent study investigated sources consumed and metabolites produced from central carbon metabolism in mixed fermentation under enological conditions with *M. pulcherrima*, *M. fructicola*, *Hanseniaspora opuntiae*, *H. uvarum*, and *S. cerevisiae* [35]. The competition for the resources seems to be the cause of strong mortality of all non-*Saccharomyces* species, particularly in mixed fermentation. This study also confirmed the importance of the evaluation of cell population dynamics and their metabolite kinetics such as glycerol and lipid uptake on the limitation of some non-*Saccharomyces* growth.

The respiratory metabolism, dependent on the overexpression of genes related to sugar consumption and cell proliferation under anaerobic and aerobic conditions in different non-*Saccharomyces* species, was investigated by Tronchoni et al. [36]. The transcriptional response to cocultivation of *S. cerevisiae* and *T. delbrueckii* was analyzed, demonstrating a metabolic interaction in HSP12 and PAU gene expression (encodes one of the two major small heat-shock proteins and fermentative growth, respectively). They found that HSP12 gene expression was stimulated in both yeasts, while PAU genes were stimulated only in *S. cerevisiae*.

In a more recent work on transcriptomic response of coculture, *M. pulcherrima* determined in *S. cerevisiae* an overexpression of the glucofermentative pathway much stronger than with the other species. In addition, in response to *M. pulcherrima*, great repression of the respiration pathway of *S. cerevisiae* was found [37]. The hypothesis formulated by the authors is a direct interaction stress response between *S. cerevisiae* and the non-*Saccharomyces* yeast. Under excess sugar conditions, respiration is inhibited while the transcription of the hexose transporters is induced, improving the fermentation rate.

The transcriptomic analysis of *S. cerevisiae*/*L. thermotolerans* mixed fermentations revealed a clear response of both yeast species to the presence of the other. In this case, it seems that genes involved in the response were related to the competition of micronutrients uptake (such as copper and iron) and those required for cell wall structure and integrity [38].

From grape to wine, the role of nutrients in winemaking is complex. The review of recent studies showed that YAN preferentially consumed by non-*Saccharomyces* yeast strains during the first stage of fermentation varies in the function of multiple factors, such as intraspecific diversity of non-*Saccharomyces* and competition with *S. cerevisiae*. However, any study carried out on the effect of nitrogen uptake during mixed fermentation depth explains why some YAN sources are preferentially consumed by non-*Saccharomyces* yeasts with respect to others. For this, the study of the gene expression involved in nitrogen and carbon regulation of *S. cerevisiae* together with transcriptional approaches, although still little explored, would be useful tools for increasing knowledge (Table 1).

## 4. Metabolic Regulation in Ethanol Reduction Using Coculture

Over recent years, there has been increasing interest in the lowering of the alcohol content in wines for human health and quality improvement. Global warming and overripened grapes due to modified consumer requests caused an alcohol concentration increase [39,40].

Among the metabolic strategies that have been applied to reduce ethanol yields, gene modification strategies on *S. cerevisiae* strains to overproduce glycerol have proven to be the most effective [41], although redox imbalances can arise, leading to an increase in undesired metabolites, including acetic acid, acetaldehyde, and acetoin, compromising the final wine quality [42]. Therefore, multiple functional genetic approaches have been explored to limit ethanol production in *S. cerevisiae*, including both the ALD6 encoding aldehyde dehydrogenase gene deletion and BDH1 encoding 2,3-butanediol dehydrogenase gene overexpression to limit acetic acid and acetoin production, respectively [43,44,45]. However, in this case, the difficult gene regulation such as the principal limitation of classical genetic engineering led to uncontrolled processes.

Recently, systemic ‘omics approaches proved to be useful in identifying specific genetic targets of modification by providing an integrated view of cell physiology, firstly describing wine yeast metabolism in detail, to understand the complex regulatory networks that occur in this organism during wine fermentation [46].

Indeed, integrated ‘omics approaches based on genomics, proteomics, and metabolomics on single *S. cerevisiae* fermentation require a broad and in-depth study of the cellular mechanisms involved in the control of the overall fermentation process and also in the reduction of ethanol as a final result. Then, following a biological strategy, the metabolic study of the coinoculation of non-*Saccharomyces* yeasts could be an alternative way to reach ethanol reduction by exploiting the reduced alcoholic fermentation efficiency of the non-*Saccharomyces* coinoculated strain. The metabolic flux distribution during fermentation differs from *S. cerevisiae* and non-*Saccharomyces* yeasts. In some non-*Saccharomyces* yeast species/strains, the diversion of alcoholic fermentation with an abundant formation of secondary compounds may in part explain the low ethanol yield. In addition, the different regulations of respirofermentative metabolism (Crabtree effect) may contribute to achieving ethanol reduction.

For these reasons, the knowledge of genes involved in the metabolic activity of non-*Saccharomyces* in mixed fermentation is a requirement to manage their use to obtain wine with low ethanol content. In this regard, Milanovic et al. [28] investigated the metabolic interaction of *Starmerella bombicola* (formerly *Candida stellata*) in an immobilized form in mixed fermentation with *S. cerevisiae.* After considerable evidence that the use of *S. bombicola* increased the glycerol content, improved the analytical profile of wine, and reduced ethanol content, it was evaluated as the metabolic mechanism through the expression with real-time RT-PCR. *S. bombicola* influenced the gene’s expression in *S. cerevisiae*: alcohol dehydrogenase (ADH1) gene expression was higher in the mixed fermentation than the pure culture differently by pyruvate decarboxylase (PDC1). This transcriptomic approach allowed us to understand that PDC1 and ADH1 genes are highly induced at the initial phase of fermentation, while at the end of the process, the expression level of PDC1 was much higher in the pure culture.

Applied studies on the use of *M. pulcherrima* in mixed fermentations showed a relevant ethanol reduction in different fermentation conditions [47,48]. A recent study on the metabolic flux of *M. pulcherrima* strains in sequential fermentation with *S. cerevisiae* showed an ethanol reduction and a higher concentration of TCA cycle byproducts (i.e., fumarate and succinate) and glycerol and lower concentrations of acetic acid [49].

Theoretical mathematical approaches such as the central composite design (CCD) and response surface methodology (RSM) were used as predictive models to investigate the potential application of non-*Saccharomyces* yeasts in ethanol reduction. For example, the cohabitation mechanisms between *S. bacillaris* (synonym *Candida zemplinina*) in combination with *S. cerevisiae* were established, both in coculture and sequential cultures, involved in ethanol reduction [50,51].

With a similar approach, Maturano and coworkers [52] recently evaluated the possibility of using mixed fermentations between a commercial starter strain of *S. cerevisiae* and *H. uvarum* and *Candida membranaefaciens,* respectively. In this case, the microbiological strategy based on the calibrated use of mixed fermentations clearly showed a significantly reduced ethanol yield when compared with *S. cerevisiae* pure culture.

Although many scientific works supported the use of mixed fermentations for ethanol reduction in wine, integrated studies on metabolic interactions between *S. cerevisiae* and non-*Saccharomyces* widely demonstrated that some associations could lead to the production of unwanted compounds such as acetic acid and ethyl acetate [53]. Non-*Saccharomyces* yeasts are characterized by respirofermentative regulatory mechanisms different from *S. cerevisiae*, and this characteristic could be used to reduce ethanol content in wine [47,54,55]. Experimental works exploring this concept are scarce due to the lack of information regarding the exact metabolic characteristics of different yeasts. The use of non-*Saccharomyces* yeasts to reduce ethanol content in wine via respiration was evaluated by Quiros et al. [54]. A screening of several non-*Saccharomyces* yeasts under aerated conditions showed that the same strains were suitable for lowering ethanol levels by respiration, in particular belonging to *M. pulcherrima*. The study of the oxidative–fermentative metabolism of different non-*Saccharomyces* yeasts strains allowed selecting *H. uvarum*, *Hanseniaspora osmophila*, *S. bacillaris*, and *C. membranifaciens* as candidates to design cocultures [56].

In this scenario (Table 2), the rich research activity around non-*Saccharomyces* wine yeasts and their metabolic traits opened new opportunities to exploit yeast metabolism with the aim of reducing the ethanol content of wines. Further investigations are needed to support the study of this aspect.

## 5. Metabolic Regulation of Volatile Compounds in Mixed Fermentation

The term “volatilome” describes the volatile organic compounds (VOCs) produced by microorganisms during alcoholic fermentation [57]. VOCs are the main responsible for the aromatic composition of wine and are closely related to yeast species, strains, and fermentation conditions [58]. During recent years, several studies were focused on the impact of non-*Saccharomyces* yeasts in mixed fermentation with *S. cerevisiae* on the aroma composition of wine, but little research has focused on the molecular mechanisms that attend in these interactions. [59,60,61,62].

During alcoholic fermentation, *S. cerevisiae* strains metabolize sugar into ethanol, fatty acids, higher alcohols, and esters, responsible for the final flavor of wine (Figure 2). The metabolic pathway involved in the production of these aroma compounds is related to the different fermentation conditions such as availability of precursors, different types of stress, the cellular redox potential and the energy status of the cell, and different types of stress [63,64,65].

The chemical and biological interactions among different yeasts during mixed fermentations underline the difficulty to understand the contribution of genes in the production of aroma compounds, making it difficult to understand every single mechanism that can influence this aspect. Several studies reported the differences in the metabolism of *S. cerevisiae* in single culture and in coculture with non-*Saccharomyces* yeasts, but a few studies have investigated the gene regulation of yeast interactions. Indeed, it is necessary to extend the genetic knowledge of non-*Saccharomyces* species with a sparsely annotated genome than a well-annotated species *S. cerevisiae*.

*T. delbrueckii* is one of the most investigated species that contributes positively to the flavor of alcoholic beverages, suggesting its profitable involvement in mixed fermentation with *S. cerevisiae* [66,67,68,69,70,71]. A recent investigation started to elucidate the metabolic differences regarding the production of aroma compounds [72]. In pure culture on synthetic grape juice medium, *T. delbrueckii* produced higher levels of ethyl propanoate, contrastingly to, *S. cerevisiae*, which exhibited a wider range of acetate and ethyl esters. This trend could be explained by transcriptome analysis, which showed the lack of the ATF1-2 gene in *T. delbrueckii* responsible for the production of acetate esters. Regarding genes related to ethyl esters, transcriptome analyses revealed different expressions between two strains, with overexpression of ETH1 in *T. delbreuckii* and a lower expression of EEB1 (biosynthesis of ethyl esters). Moreover, the low production of higher alcohols in *T. delbreuckii* is related to the catabolism of branched-chain amino acids (BCAAs; leucine, valine, and isoleucine) through the Ehrlich pathway and regulated by BAT1, BAT2, and BAP2 genes that are not transcribed in *T. delbreuckii*. Agarbati and colleagues [73] assessed the possible employment of *T. delbrueckii* to produce Verdicchio wine with reduced sulfites through sequential fermentation with *S. cerevisiae* to determine an increase in aroma compounds such as phenyl ethyl acetate and ethyl hexanoate. The modality of inoculation also affects the volatile thiol production in *T. delbrueckii* during alcoholic fermentation. In sequential fermentation with *S. cerevisiae*, higher levels of 3-sulfanylhexan-1-ol (3SH) and its acetate ester were observed in comparison with pure culture, highlighting a synergistic interaction between the two species. To understand this result, the metabolism of the precursors responsible for these aromatic compounds, glutathionylated conjugate precursor (Glut-3SH) and cysteinylated conjugate precursor (Cys-3SH), were analyzed [74]. The results showed that *S. cerevisiae* metabolized the two precursor forms, while *T. delbrueckii* was able to metabolize the glutathionylated precursor. Consequently, the presence of *T. delbrueckii* during mixed fermentation led to an increase in Glut-3SH degradation and Cys-3SH production. This overproduction was dependent on the *T. delbrueckii* biomass. In sequential culture, thus favoring *T. delbrueckii* development, the higher availability of Cys-3SH throughout AF resulted in more abundant 3SH and 3SHA production by *S. cerevisiae. H. vinae* also is a non-*Saccharomyces* species used in mixed fermentation to improve the aromatic profile of wines by the production of 2-phenylethyl acetate, acetate esters, medium-chain fatty acid ethyl esters, benzenoids, and terpenes [75]. The study of the genome sequencing of *H. vinae* in sequential fermentation in synthetic medium revealed that the increase in 2-phenylethyl acetate and phenylpropanoids was linked to gene duplications of aromatic amino acid aminotransferases encoded by ARO8 and ARO9 genes and phenylpyruvate decarboxylases encoded by ARO10 [76]. The absence of the branched-chain amino acid transaminases (BAT2) and acyl-coenzyme A (acyl-CoA)/ethanol O-acyltransferases (EEB1) genes in *H. vineae* reduced the production of branched-chain higher alcohols, fatty acids, and ethyl esters, respectively.

Another important group of volatile compounds that affects wine flavor comprises sulfur compounds. It has long been shown that H_2_S can be produced by yeasts using elemental sulfur with reducing compounds [77]. H_2_S is converted to cysteine, methionine, and glutathione via a series of enzyme-catalyzed reactions. During the growth phase of yeast, sulfur amino acids are used in protein synthesis, while later in the fermentation process, they can be excreted from the cell and appear in the finished wine. When cellular nitrogen levels are limited, intracellular metabolic pathways can lead to the production of unpleasant H_2_S. Therefore, in the issue of mixed fermentation, the understanding of the metabolic mechanisms that lead to the formation of these compounds is essential to avoid high levels of H_2_S by non-*Saccharomyces* strong H_2_S producers. However, very few reports are available in the literature concerning the production of sulfur compounds by non-*Saccharomyces* yeasts. Moreira et al. [78,79] studied the effect of pure and mixed cultures of *H. guilliermondii*, *H. uvarum*, and *S. cerevisiae* in sulfur production in wine. The levels of heavy sulfur compounds in mixed cultures with *S. cerevisiae* of both apiculate yeasts were low and like those obtained in a pure culture of *S. cerevisiae*, highlighting the positive role of mixed fermentation.

Transcriptional analysis on non-*Saccharomyces* and *S. cerevisiae* could be a suitable strategy to understand and set up strategies to modulate the transcriptional response of specific genes that are linked to the production of aroma compounds of both *S. cerevisiae* and non-*Saccharomyces* yeasts (Table 3). Indeed, a recent review, focusing on the strategies for enhancing aroma production by yeast during wine fermentation, highlighted the relevant role of selected non-*Saccharomyces* yeasts in mixed fermentation to expand the aroma profiles of the wines [80]. About this, it is necessary to improve genetic information about non-*Saccharomyces* yeasts to delineate the genes and metabolic pathways involved in the production of aromatic compounds in mixed fermentation. In this regard, a recent work developed genetic tools for *H. uvarum* to reduce ethyl acetate through the disruption of the HuATF1 genes that encodes a putative alcohol acetyltransferase [81].

## 6. Yeast-Yeast Interaction and Antimicrobial Activity

Among the wide applications of non-*Saccharomyces*/*S. cerevisiae* mixed fermentations in winemaking, the control of undesired microorganisms is one of the actual and promising features. Indeed, there is a growing global interest in biocontrol procedures in foods and in beverages, based on the environmentally sustainable trend. The regulation of the growth of undesired microorganisms could be exploited through antagonistic action, cell-to-cell contact, or through the production of antimicrobial compounds such as mycocins, small peptides, or extracellular vesicles (Table 4). Mycocins are clearly involved in the yeast–yeast interactions in mixed fermentations. Several works have focused on the study of non-*Saccharomyces* strains able to counteract the development of *Brettanomyces* spp., a relevant dangerous yeast in the cellar [82,83,84].

The mycocin Kpkt produced by *Tetrapisispora phaffii* was first described as antispoilage yeast [85]. Kpkt acts through a specific β-glucanase activity causing irreversible modifications on the cell wall structure, and it is codified by the TpBGL2 chromosomal gene [86,87,88]. The recombinant toxin (rKpkt) was recently obtained by transferring the Kpkt-coding gene in *Komagataella phaffii* (formerly *Pichia pastoris*) [89]. The recombinant Kpkt, when expressed in *K. phaffii*, displayed a wider spectrum of action than in its native yeast [90], reinforcing the idea of the possible application of mycocins and/or killer yeasts in the food and beverages industries. *T. phaffii* was used in mixed fermentations at the prefermentative stage to control wild yeasts such as *Hanseniaspora*, *Zygosaccharomyces*, and *Saccharomycodes* as substitutes of sulfur dioxide [91].

Most studies on mycocins are focused on counteracting the development of the *Dekkera*/*Brettanomyces* wine-spoilage yeasts. In this context, Pikt and Kwkt mycocins, produced by *Pichia anomala* and *Kluyveromyces wickerhamii*, respectively [92], could be used to counteract *Dekkera*/*Brettanomyces*. In particular, Pikt is a ubiquitin-like protein of about 8 kDa able to interact with β-1,6-glucan of the cell wall of sensitive yeasts [93]. Kwkt, a protein of about 72 kDa of molecular mass, without any glycosyl residue [94] and β-1,6-glucosidase activity, seems to be involved in blocking the cell cycle function of sensitive yeasts [95]. Through fluorescence techniques of cell death due to alterations in yeast cell permeability and cell metabolism (cytofluorimetric evaluation), it was demonstrated that both Pikt and Kwkt caused the irreversible death of this yeast.

Differently, sulfur dioxide induced a viable but noncultivable (VBNC) state of *Brettanomyces* with the consequent recovery of yeasts when fresh medium was replaced [95].

*P. membranifaciens* yeast was also described as a killer yeast able to produce two mycocins: PMKT and PMKT2. The first mycocin binds linear (1→6)-β-d-glucans in the cell wall and Cwp2p plasma membrane receptor of sensitive yeasts, leading to alterations in ionic exchange via the plasma membrane [96]. PMKT induces the activation of genes such as CTT1, HSP12, GPD1, GPP2, TRK2, PDR12, ENA1, SCH9, HAL9, YAP1, XBP1, and STL1, involved in resisting osmotic shock [97,98]. PMKT2, a protein with an apparent molecular mass of 30 kDa, binds mannoproteins and induces cell cycle blockage in an early S-phase of sensitive yeasts, and stimulates markers of cellular apoptosis such as the cytochrome c release, DNA strand breaks, metacaspase activation, and production of reactive oxygen species at a low dose [98,99].

The killer activity of *P. membranifaciens* was exploited in mixed fermentation with *S. cerevisiae* and *Brettanomyces bruxellensis* (inoculum ratio of 1:1) in mixed wine fermentation. *P. membranifaciens* inhibited *B. bruxellensis* growth [100] without deleterious effects on the fermentation performance of *S. cerevisiae*.

Other mycocins active against *B. bruxellensis* produced by *Candida pyralidae* and denominated CpKT1 and CpKT2 (both about 50 kDa) were partially characterized: both were active and stable at pH 3.5–4.5 and in a temperature range between 15 and 25 °C. Furthermore, their killer activity was not reduced/suppressed by the ethanol and sugars presence, highlighting their compatibility with winemaking conditions [101]. Subsequently, CpKT1 was described as a protein with β-glucanase activity able to lead to cell membrane and cell wall damage. Both CpKT1 and *C. pyralidae* yeast were used in mixed fermentations in red grape juice containing *B. bruxellensis* with a consequent decrease in spoilage yeast concentration [102]. A strain of *Wickerhamomyces anomalus* was proposed as a valid biocontrol agent against *Brettanomyces*/*Dekkera* spp. [103]. The killer activity of *W. anomalus* is expressed through the release of KTCf20, a mycocin that binds β-1,3- and β-1,6-glucans of sensitive yeasts’ cell walls. This mycocin was able to counteract the growth of *Brettanomyces*/*Dekkera*, *P. guilliermondii*, and *P. membranifaciens*. Moreover, they showed that *W. anomalus* did not negatively affect the presence of *S. cerevisiae* strains in mixed fermentations. Another mycocin produced by another strain of *W. anomalus* and active against *B. bruxellensis* was also reported by Comitini et al. [104]. The mycocin denominated WA18 is a protein with 99% UDP-glycosyltransferase protein identity and branched beta-glucans, representing the first mycocin receptors on the surface of the sensitive yeasts. In accordance with de Ullivarri and coworkers [103], they confirmed the compatibility of *W. anomalus* strain in mixed fermentation with *S. cerevisiae* yeast.

It was also described that *T. delbrueckii* is able to release the TdKT mycocin (>30 kDa) with glucanase and chitinase activities. This mycocin is stable in wine environmental conditions, and pustulan and chitin seem to be the first toxin targets in the cell wall, causing cell wall damage, necrosis/apoptotic cell death of sensitive yeasts such as *B. bruxellensis*, and other potential wine-spoilage yeasts [105]. Moreover, Ramírez et al. [106] isolated and selected wine *T. delbrueckii* strains producing killer toxin Kbarr-1. This toxin is encoded by a dsRNA, TdV-Mbarr-1, structurally like M dsRNAs of *S. cerevisiae*, which both seem to be evolutionarily related [107].

Until now, non-*Saccharomyces* yeasts were able to exploit their killer activity through the production of mycocins active against other non-*Saccharomyces* yeasts. However, it was also described as the application of *Saccharomyces* spp. killer strains/mycocins versus grape/wine-spoilage yeasts. Recently, a strain of *Saccharomyces eubayanus* was studied able to secrete a mycocin (SeKT) with a molecular mass of about 70 kDa that reduces the levels of volatile phenols produced by wine-spoilage yeasts such as *B. bruxellensis*, *Meyerozyma guilliermondii*, *P. membranifaciens*, and *Pichia manshurica*. This mycocin acts through β-glucanase and chitinase activities, leading to cell wall disruption [108,109].

In the yeast–yeast interactions in wine fermentation, *Saccharomyces* yeasts play an important role in antimicrobial activity through the production of mycocins and secretion of extracellular antimicrobial peptides (AMPs), recently named “saccharomycin”, derived from the glycolytic enzyme glyceraldehyde 3-phosphate dehydrogenase (GAPDH), with a wide range of action. AMPs are low-molecular-weight proteins active against bacteria, viruses, and fungi. Only a limited number of articles have reported AMPs secreted by yeast with activity against non-*Saccharomyces* wine-related strains. The most studied are AMPs from *S. cerevisiae* with activity against a variety of wine-related yeasts [110,111]. These peptides are C-terminal fragments of glyceraldehyde 3-phosphate dehydrogenase (GAPDH) with a molecular mass close to 1.6 kDa, with interesting antifungal properties against *H. guilliermondii*, *Kluyveromyces marxianus*, *L. thermotolerans*, and *T. delbrueckii* [112]. The membrane functionality seems to have a key role in the antimicrobial activity of AMPs by the production of amphiphilic structures that interact with the exposed receptors (carbohydrate molecules). This was recently demonstrated by Caldeira et al. [113] by structural characterization of saccharomycin carried out by nuclear magnetic resonance (NMR). Moreover, Pena and Gang [114] and Pena et al. [115] found AMPs, of about 5 kDa produced by *Candida intermedia*, able to stimulate the production of reactive oxygen species (ROS) in sensitive yeasts. These AMPs showed an effective antimicrobial activity in wine against two strains of *P. guilliermondii* and *B. bruxellensis*, considered an excellent application of a new strategy for the biocontrol of spoilage yeasts.

Among yeast–yeast interactions, the culturability loss of non-*Saccharomyces* yeasts has received growing interest due to new findings on the role of excreted compounds in the interaction between *Saccharomyces* and non-*Saccharomyces* yeasts [116,117,118]. Branco et al. [111] showed that VBNC status was related to interaction through secreted compounds. The loss of culturability was investigated in more non-*Saccharomyces* wine yeasts as starters in mixed fermentation with *S. cerevisiae* [119] with the aim of understanding their final impact on wine quality. Another work [120] confirmed that some metabolites produced by *S. cerevisiae* played the main role in the decreased cultivability of the other *Saccharomyces* and non-*Saccharomyces* yeasts, indicating that the interactions are species and strain specific. For example, it was found that the main cause for the lack of cultivability of *H. uvarum* did not seem to be due to cell-to-cell contact but rather compounds related to fermentations such as ethanol and/or certain metabolites secreted by *S. cerevisiae*. Another possible mechanism of interaction in mixed fermentations is the possible involvement of extracellular vehicles (EVs). In this regard, recent work on exo-proteome in pure and mixed fermentations non-*Saccharomyces* and *S. cerevisiae* investigated proteomic analysis of EV-enriched fractions from six different species. Results showed a wide diversity of proteins secreted, indicating the presence of interactions and the possible involvement of EVs [121,122]. The EV-enriched fractions from different species such as *S. cerevisiae*, *T. delbrueckii*, and *L. thermotolerans* showed enrichment in glycolytic enzymes and cell-wall-related proteins and particularly the enzyme exo-1,3-β-glucanase. However, this protein was not involved in the here-observed negative impact of *T. delbrueckii* extracellular fractions on the growth of other yeast species. These findings suggest that EVs may play a role in fungal interactions during wine fermentation and other aspects of wine yeast biology.

Another antimicrobial interaction in wine fermentation is the release of extracellular compounds such as peptides, acids, and other small molecules. *M. pulcherrima* exerts its antagonistic action through pulcherriminic acid (precursor of pulcherrimin pigment) production, depleting the iron present in the medium and consequently not making it available for other yeasts. *M. pulcherrima*, when inoculated in coculture with *H. guilliermondii*, *P. membranifaciens*, and *B. bruxellensis*, exerted an antimicrobial action, while in coculture with *S. cerevisiae*, it did not show antimicrobial activity [123]. More recently, Kántor et al. [124] demonstrated the in vitro antimicrobial action of *M. pulcherrima* against *Candida* spp. and *P. manshurica* grape/wine-related yeasts.

The death of some wine-related yeast species during mixed-culture fermentations has always been attributed to their inability to survive in the presence of selective growth factors. However, it was shown that mechanisms based on cell-to-cell contact were able to influence yeast population dynamics during wine fermentation [125]. Double-compartment fermentation system used for *L. thermotolerans*/*S. cerevisiae* mix fermentations showed a reduction in the death rate of *L. thermotolerans* than in noncompartmentalized mixed-culture fermentations, despite the two fermentation systems showing comparable amounts of the antimicrobial peptidic fraction [126]. Similarly, *S. bacillaris* died earlier when tested in mixed fermentation with *S. cerevisiae* using the flask compared to when both yeast species were kept physically separate [127]. These results highlight that the death of these non-*Saccharomyces* yeasts seems to be not caused by a nutrient limitation or growth-inhibitory compound accumulation but rather by cell-to-cell contact mechanisms. Moreover, the cell-to-cell contact mechanism seems also to influence the metabolic behavior of the yeast strains with the consequent production (or not) of specific chemical compounds, affecting the aroma characteristics of wine [128].

## 7. Conclusions

Controlled mixed fermentations undoubtedly have a positive impact on the analytical and sensory characteristics of wines and may have a positive role in the biocontrol procedures to reduce the use of chemical compounds. On the other hand, the behavior of the coculture cannot be accurately predicted, and fermentations carried out in a reproducible manner remains a big challenge.

For this, further knowledge is needed on yeast–yeast interactions and in relation to the environmental factors, as the molecular mechanisms of these interactions are largely unknown. While *S. cerevisiae* is genetically well characterized, genome information on non-*Saccharomyces* is still in an early stage, and further investigations should be carried out to obtain a more clearer picture in identifying certain key genes and pathways. Figure 3 summarizes the overviewed functions of the use of non-*Saccharomyces* wine yeasts by means of SWOT analysis. where each item assesses the strengths, weaknesses, opportunities, and threats of this issue.

## Figures and Tables

**Figure 1 ijms-22-07754-f001:**
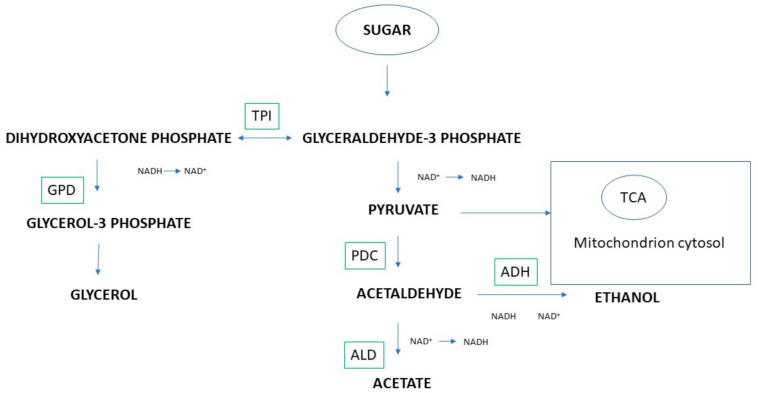
Overview of sugar catabolism in yeast cells during alcoholic fermentation. GPD: glycerol-3-phosphate dehydrogenase; PDC: pyruvate decarboxylase; ADH: alcohol dehydrogenase; ALD: aldehyde dehydrogenase; TPI: triosephosphate isomerase. Pyruvate originating is partially shuttled to the mitochondrion and incorporated into the tricarboxylic acid (TCA) cycle.

**Figure 2 ijms-22-07754-f002:**
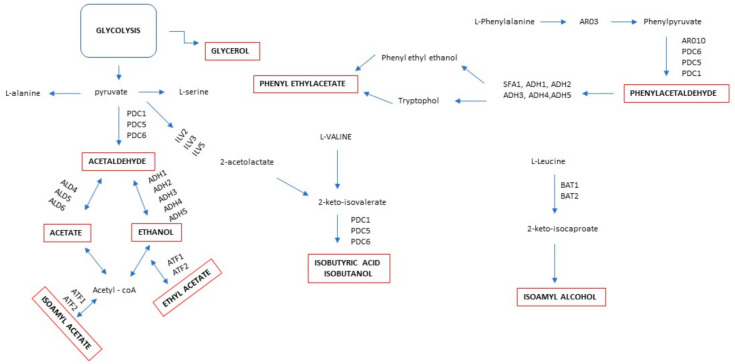
The main metabolic pathway involved in the production of aroma compounds. PDC1, PDC5, and PDC6 genes coding pyruvate decarboxylase; ALD4, ALD5, and ALD6 genes coding aldehyde dehydrogenase; ADH1, ADH2, ADH3, ADH4, and ADH5 genes coding alcohol dehydrogenase; ATF1 and ATF2 genes coding alcohol acetyl transferase; ILV2, ILV3, and ILV5 genes coding acetolactate synthase; AR03 and AR010 genes coding phenyl pyruvate and phenyl acetaldehyde; SFA1 synonyms of ADH5; BAT1 and BAT2 gene coding the mitochondrial-targeting signal.

**Figure 3 ijms-22-07754-f003:**
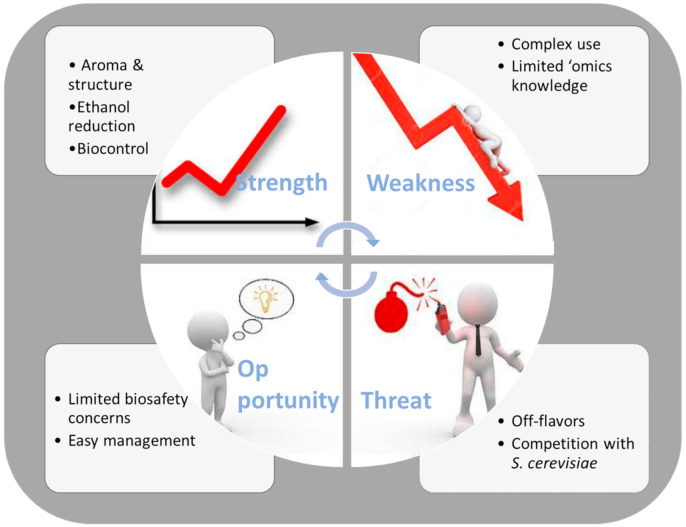
SWOT diagram, summarizing the strengths (S), weaknesses (W), opportunities (O), and threats (T) of the current management of non-Saccharomyces yeasts.

**Table 1 ijms-22-07754-t001:** The main metabolic response in yeast–yeast interactions.

Species	Metabolic Pathway/Gene Regulation	Fermentation Condition	References
*S. bacillaris*	Histidine, methionine, threonine, and tyrosine were not consumed; cysteine preferred the nitrogen source	Mixed fermentation	[21,30,31,32]
	Low activity of the acetaldehyde pathway		
*M. pulcherrima*	Aspartic acid assimilated slowly, cysteine preferred nitrogen source	Mixed fermentation	[21]
*P. membranifaciens*	Glutamine was not assimilated; cysteine preferred the nitrogen source	Mixed fermentation	[21]
*S. cerevisiae*	Genes related to the biosynthesis of vitamins were upregulated while genes involved in the uptake and biosynthesis of amino acids were downregulated	Mixed fermentation with *H. guilliermondii*	[17]
	Stimulation of nitrogen metabolism	Mixed fermentation with *C. sake* and *H. uvarum*	[22]
	Reduced the ammonium effects at fermentation kinetics; stimulated PAU and HSP12 genes	Mixed fermentation with *T. delbrueckii*	[23,36]
	High expression of pyruvate decarboxylase (PDC1) and (PDC5), acetaldehyde dehydrogenase (ALD6), alcohol dehydrogenase (ADH1) and glycerol-3-phosphate dehydrogenase (PDC1); overexpression of the glucofermentative pathway and great repression of the respiration pathway	Mixed fermentation with *M. pulcherrima*	[33,37]
	Accelerated the nutrient uptake	Mixed fermentation with *S. kudriavzevii*	[34]
*H. vineae*	Nitrogen catabolism repression	Mixed fermentation synthetic must	[24]
*T. delbrueckii*	Stimulation HSP12 gene (encodes one of the two major small heat-shock proteins)	Mixed fermentation	[36]

**Table 2 ijms-22-07754-t002:** Metabolic regulation in ethanol reduction in different wine yeasts.

Species	Metabolic Pathway/Gene Regulation	Fermentation Condition	References
*S. cerevisiae*	Gene modification: aldehyde dehydrogenase (ALD6) gene deletion; 2,3-butanediol dehydrogenase (BDH1) gene overexpression	Pure culture	[43,44,45]
*S. bombicola*	Alcohol dehydrogenase (ADH1) gene expression increased; pyruvate decarboxylase (PDC1) decrease	Immobilized form in mixed fermentation	[28]
*M. pulcherrima*	higher concentrations of TCA cycle byproducts (i.e., fumarate and succinate)	Sequential culture	[49]
*H. uvarum* and *C. membranaefaciens*	calibrate the use of mixed fermentation clearly showed a significantly reduced ethanol yield	Mixed fermentation	[52]
*M. pulcherrima*	Metabolism respirofermentative	Pure culture	[54]
*H. uvarum*, *H. osmophila*, *S. bacillaris* and *C. membranafaciens*	Oxidative–fermentative metabolism	Coculture	[56]

**Table 3 ijms-22-07754-t003:** Metabolic regulation of volatile compounds in different wine yeasts.

Yeast Species	Volatile Compounds	Gene/Metabolic Regulation	Fermentation Condition	References
*T. delbrueckii*	Increase ethyl propanoate	Lack of the ATF1-2 gene	pure culture on synthetic grape juice medium	[72]
	Ethyl esters	Overexpression EHT1 gene and low expression of EEB1 gene		
	Low production higher alcohols	Catabolism of branched-chain amino acids (BCAAs; leucine, valine, and isoleucine) regulated by BAT1, BAT2, and BAP2 genes that are not transcribed in *T. delbreuckii*		
	Higher levels of 3-sulfanylhexan-1-ol and acetate ester	Glut-3SH (glutathionylated conjugate precursor) and Cys-3SH (cysteinylated conjugate precursor)	Sequential fermentation	[74]
*H. vinae*	2-phenylethyl acetate, acetate esters, medium-chain fatty acid ethyl esters, benzenoids, and terpenes	ARO8 and ARO9 genes coding aromatic amino acid aminotransferases; ARO10 gene coding phenylpyruvate decarboxylases.	Mixed fermentation	[75]
	Reduced production of branched-chain higher alcohols, fatty acids, and ethyl esters	Absence of the branched-chain amino acid transaminases (BAT2) and acyl-coenzyme A (acyl-CoA)/ethanol O-acyltransferases (EEB1) genes	Sequential fermentation in synthetic grape juice	[76]
*H. uvarum*	Reduce ethyl acetate	Disruption of the HuATF1 genes	Mixed fermentation	[81]

**Table 4 ijms-22-07754-t004:** Main antimicrobial activity in yeast–yeast interactions.

Antimicrobial Features	Yeast Specie	Antimicrobial Activity	Wine Management	Gene Regulation	References
Mycocins	*Tetrapisispora phaffii*	Kpkt_β-glucanase activity	Prefermentative stage of mixed wine fermentation	*TpBGL2* chromosomal gene	[91,123]
	*Pichia anomala*	Pikt_ubiquitin-like protein	Biocontrol agent against wine-spoilage yeasts	-	[92,93]
	*Kluyveromyces wickerhamii*	Kwkt_ β-1,6-glucosidase activity	Biocontrol agent against wine-spoilage yeasts	-	[92,95]
	*Pichia membranifaciens*	-	Mixed wine fermentation	*CTT1*, *HSP12*, *GPD1*, *GPP2*, *TRK2*, *PDR12*, *ENA1*, *SCH9*, *HAL9*, *YAP1*, *XBP1*, and *STL1* genes involved in osmotic shock	[96,97,98,100]
	*Pichia membranifaciens*	-	Mixed wine fermentation	Stimulates markers of cellular apoptosis	[98,99,100]
	*Candida pyralidae*	CpKT1_β-glucanase activity	Mixed wine fermentation	-	[102]
	*Wickerhamomyces anomalus*	WA18_9UDP-glycosyltransferase	Mixed wine fermentation	-	[104]
	*Torulaspora delbrueckii*	TdKT_glucanase and chitinase activities	Biocontrol agent against wine-spoilage yeasts	-	[105]
	*Torulaspora delbrueckii*	-	Biocontrol agent against wine-spoilage yeasts	Encoded by TdV-Mbarr-1 dsRNA	[106,107]
	*Saccharomyces eubayanus*	SeKT_β-glucanase and chitinase activities	Biocontrol agent against wine-spoilage yeasts	-	[108,109]
Extracellular antimicrobial peptides	*Saccharomyces cerevisiae*	AMPs_fragments of glyceraldehyde 3-phosphate dehydrogenase	Mixed wine fermentation	-	[112]
	*Candida intermedia*	AMPs_stimulate reactive oxygen species (ROS)	Biocontrol agent against wine-spoilage yeasts	-	[114,115]
Extracellular vesicles and compounds	*Saccharomyces cerevisiae*	Evs_glycolytic enzymes	Biocontrol agent	-	[122]
	*Torulaspora delbrueckii*	Evs_glycolytic enzymes	Biocontrol agent	-	[37]
	*Lachancea thermotolerans*	Evs_glycolytic enzymes	Biocontrol agent	-	[122]
	*Metschnikowia pulcherrima*	Pulcherriminic acid	Biocontrol agent against wine-spoilage yeasts	-	[123,124]
Cell-to-cell contact	*Lachancea thermotolerans*/*Saccharomyces cerevisiae*	Cell-to-cell contact	Mixed wine fermentation	-	[126]
	*Starmerella bacillaris*/*Saccharomyces cerevisiae*	Cell-to-cell contact	Mixed wine fermentation	-	[127]

(-): unavailable information.

## Data Availability

Not applicable.

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
