# Peer review of "Yeast Interactions and Molecular Mechanisms in Wine Fermentation: A Comprehensive Review"

_ijms, 2021, doi:10.3390/ijms22147754_

Round 1

Reviewer 1 Report

  1. Despite the fact that in the past year lots of authors focused on the use of non-Saccharomyces, the present manuscript has focused on the recent advanced methodological approach in the field, which is very welcome.
  2. The number of the rows is not present in the entire manuscript, making difficult to refer to an issue or to make an observation. The comments/observations are added directly on the manuscript
  3. Try to increase the clarity of figure 1, figure 2

Author Response

Reviewer 1 (following the suggestions and/or correction made in pdf file)

Despite the fact that in the past year lots of authors focused on the use of non-Saccharomyces, the present manuscript has focused on the recent advanced methodological approach in the field, which is very welcome.

The number of the rows is not present in the entire manuscript, making difficult to refer to an issue or to make an observation. The comments/observations are added directly on the manuscript

Try to increase the clarity of figure 1, figure 2

ANSWER to Reviewer 1:

Thank you for your suggestions. We improved and clarified the manuscript following your suggestions. You can find the corrections highlighted in yellow in the manuscript. We added the rows in the entiere manuscript.

Regarding to the last paragraph in the section 3, the sentence was carried out to summaries the concepts reported above. For this reason, some already cited concept are corroborated. For this, we prefer to maintain the original form.

Reading to the figure1 and 2, we added some general information in figure (Fig. 1) caption and details related to genes reported (Fig. 2).

Regarding to the references, the double number was delete.

Reviewer 2 Report

Ref: International Journal of Molecular Sciences

ID: ijms-1293657

General comments:

The review described about interactions among wine yeasts, particularly during fermentations carried out by mixed starter cultures of two or more yeast species.

The manuscript is well written, the description of the main topics that are influenced by the yeast-yeast interactions was well presented supported by an adequate literature. In my opinion, the work is acceptable for publication after the following minor corrections:

- Section 2, page 2: in the last paragraph the methodologies FT-ICR-MS and LC-MS/MS are described but no bibliography is cited.

- Section 3, page 4: at the end of the first paragraph correct Barbosa and co-workers.

- Section 5, page 9:

  1. a) correct the third sentence by “During the last years, …”
  2. b) revise the last sentence that is repeated, delete “availability of precursors, …”
  3. c) add the definition of abbreviations in the caption of the Figure 2, for example BAT: branched-chain aminoacid transaminases

- Please verify the taxonomic name form of Hanseniaspora vineae.

- At the end of page 10, please replace high H2S producers by strong H2S producers.

- Page 15 Line 35: please replace Ullivarri by Fernandez de Ullivarri and co-workers and correct in the list of references (103).

- Page 16 Line 76: delete viable but not-culturable, the abbreviation was defined above.

- Conclusions, Lines 124-125: please rephrase the sentence adding “The way in which the co-culture…” or “The behavior of the co-culture cannot …”

Author Response

General comments:

The review described about interactions among wine yeasts, particularly during fermentations carried out by mixed starter cultures of two or more yeast species.

The manuscript is well written, the description of the main topics that are influenced by the yeast-yeast interactions was well presented supported by an adequate literature. In my opinion, the work is acceptable for publication after the following minor corrections:

- Section 2, page 2: in the last paragraph the methodologies FT-ICR-MS and LC-MS/MS are described but no bibliography is cited.

ANSWER to Reviewer 2:

We added the required reference in the text

- Section 3, page 4: at the end of the first paragraph correct Barbosa and co-workers.

ANSWER: corrected in the text

- Section 5, page 9:

  1. a) correct the third sentence by “During the last years, …”
  2. b) revise the last sentence that is repeated, delete “availability of precursors, …”
  3. c) add the definition of abbreviations in the caption of the Figure 2, for example BAT: branched-chain aminoacid transaminases

ANSWER: we corrected in the text.

- Please verify the taxonomic name form of Hanseniaspora vineae.

ANSWER: we verified in this link https://www.ncbi.nlm.nih.gov/Taxonomy/Browser/wwwtax.cgi?id=56409

- At the end of page 10, please replace high H2S producers by strong H2S producers.

ANSWER: we corrected in the text.

- Page 15 Line 35: please replace Ullivarri by Fernandez de Ullivarri and co-workers and correct in the list of references (103).

ANSWER: we corrected in the text.

- Page 16 Line 76: delete viable but not-culturable, the abbreviation was defined above.

ANSWER: we corrected in the text.

- Conclusions, Lines 124-125: please rephrase the sentence adding “The way in which the co-culture…” or “The behavior of the co-culture cannot …”

ANSWER: we corrected in the text.